# Application of Hybrid Swarming Algorithm on a UAV Regional Logistics Distribution

**DOI:** 10.3390/biomimetics8010096

**Published:** 2023-02-27

**Authors:** Yi Zhang, Hongda Yu

**Affiliations:** 1College of Electrical and Computer Science, Jilin Jianzhu University, Changchun 130000, China; 2Key Laboratory for Comprehensive Energy Saving of Cold Regions Architecture of Ministry of Education, Jilin Jianzhu University, Changchun 130118, China

**Keywords:** COVID-19, Physarum Polycephalum algorithm, genetic algorithm, Van der Waals force, UAV logistics and distribution

## Abstract

This paper proposes a hybrid algorithm based on the ant colony and Physarum Polycephalum algorithms. The positive feedback mechanism is used to find the globally optimal path. The crossover and mutation operations of the genetic algorithm are introduced into the path search mechanism for the first time. The Van der Waals force is applied to the pheromone updating mechanism. Simulation results show that the improved algorithm has advantages in quality and speed of solution compared with other mainstream algorithms. This paper provides fast and accurate route methods for solving the Traveling Salesman Problem first and a delivery scheme is also presented for UAVs to realize “contactless delivery” to users in the Changchun Mingzhu District during the COVID-19 epidemic, which confirms the practicability and robustness of the algorithm.

## 1. Introduction

The pandemic has posed unprecedented challenges and pressures to the provision of essential supplies that support protecting people’s livelihoods and the stability of society. Contactless delivery and uncrewed transportation are increasingly challenging, especially in closed or partially closed areas, with disrupted transportation routes and blocked supply chains. The advancement of drone technology makes it possible for drone delivery to gradually become part of daily life and improve user experience and quality of life. Drone delivery can provide fast, efficient, and safe delivery services for medical supplies, daily necessities, and other goods, reducing the risk of virus transmission and supporting epidemic control efforts. Uncrewed Aerial Vehicle (UAV) technology is poised to play a vital role in helping people and communities overcome the challenges posed by the epidemic. This paper provides a reliable and efficient implementation plan for this “contactless interaction.” (Added background.) The “contactless distribution” has become a research focus of scholars during the pandemic of COVID-19 around the world. The goal is to achieve a fast and efficient contactless distribution scheme between the distribution center and the user. Victoria Lyon [1] provides a regional logistics distribution to avoid direct person-to-person contact that is effective for during COVID-19. Golabi M [2] proposed using UAVs to carry out rescue work; focusing on rescue work after an earthquake, taking the Tehran earthquake as an example. Huo Da [3] presents a new and efficient “last kilometer” distribution scheme. In addition, Li Y [4] used the simulated annealing algorithm to focus on the logistics of uncrewed aerial vehicle (UAV) distribution in urban environments for contactless distribution. Li Y proposes a multiobjective decision-making method and uses a Variable Neighborhood Search (VNS) framework to generate an approximate optimal solution for this problem. Manuel Patchou [5] presents mobile robotic systems inherently for minimizing human interaction during parcel delivery following the unexpected outbreak of the COVID-19 pandemic. 

The model of the regional distribution problem is the traditional Traveling Salesman Problem (TSP). Many scholars have obtained good research results on the TSP. Some scholars use bionic optimization algorithms to improve the solving accuracy. The ant colony algorithm is a bionic iterative search algorithm inspired by the foraging behavior of ant colonies in nature. The ants can always find the shortest path between food and the nest during foraging. The ant colony will quickly adjust its path to form a new shortest path when the original obstacle is removed, or a new obstacle is added. The main reason is that ant colonies can release and detect “pheromones” when finding food. This chemical can guide ant colonies as a medium for communication between ant colonies. Other individuals can just read the route by identifying pheromones to obtain current information. With trial and error, and the exploration of ant colonies, they gradually find the best route to find food. (Added basic method.) He yi [6] proposed adjusting the ACA parameters and applying the chaos theory to the adaptive adjustment of parameters. Finally, the algorithm can jump out of the local optimal in the iterative process. Cheng biyun [7] presents a concept of the excellent coefficient. She applied it to discrete Particle Swarm Optimization (PSO) to solve the TSP problem. The all-around performance of the algorithm was improved correspondingly. Min Kexue [8] and other researchers mixed the PSO in ACA for optimization processing. According to the PSO’s characteristics, each ant’s five parameters are taken as a particle of the PSO. A particle swarm is used to optimize each ant colony’s development and exploration ability to avoid the ant colony’s premature convergence to the optimal local solution. Additionally, several heuristic optimization algorithms for solving the TSP, such as the genetic algorithm, Particle Swarm Algorithm, Artificial Fish Swarm Algorithm, Grey Wolf Algorithms, and Frog-Leap Algorithms. As a robust search algorithm, the genetic algorithm borrows concepts from biology, such as chromosomes and genes, and mimics natural inheritance and evolution. The ant colony algorithm still differs in algorithm mechanisms, implementation forms, and limitations in solving specific problems. The ant colony algorithm has a significant advantage over other heuristic algorithms in solving the TSP. However, it usually requires a long search time, may become stuck, and is sensitive to the initialization parameters. (Comparison with other heuristics).

The above research used parameter adjustment to optimize the results. The obtained results also show that the algorithm is effective after parameter adjustment. The effect of adjusting parameters on improving the accuracy of the solution needs to be made apparent. However, adjusting parameters plays a particular role in the algorithm jumping out of the local optimal. So much research focuses on using the algorithm and other algorithms’ advantages to improve accuracy. W. Junqiang [9] proposed a hybrid ant colony algorithm (HACO) by aiming for the traditional ant colony algorithm. Experiments show that the hybrid algorithm is superior to the traditional ant colony algorithm in every aspect. H. Qian [10] proposed optimizing the ant colony algorithm, combining the maximum and minimum ACA with PSO (ACO-PSO) to solve the Traveling Salesman Problem. H. Min [11] presents a hybrid ACA in which ant colonies of each generation are crossed and mutated. New individuals with a specified probability accept according to the Metropolis criterion of the simulated annealing algorithm. This mechanism is conducive to the algorithm’s convergence but not conducive to the diversity of the algorithm search. A typical ACO combined with an NPI strategy algorithm is proposed by S. Gao [12] called the NPI-ACS algorithm. Simulation experiments prove the flexibility of this algorithm.

Dewantoro, R.W. [13] proposes using an ant colony algorithm and taboo search algorithm as the local search to solve the TSP. Comparative experiments show that the proposed algorithm has a better route and shorter running time than the ant colony algorithm. Gabhane, J.P [14] presents a hybrid algorithm with the ant colony algorithm and the taboo search algorithm (TS) to design a novel algorithm (ACOTS) to solve the scheduling problem of cloud user workloads. The proposed algorithm saves 30% of time costs compared with the genetic algorithm, Particle Swarm Algorithm, ant colony algorithm, and Tabu Search Algorithm. Qamar, M.S. [15] presents a novel Best-Worst Ant System (BWAS) based algorithm to settle the Traveling Salesman Problem (TSP). The work proposed a BWAS-based incorporated arrangement as a high-level type of ACO to upgrade the exhibition of the TSP arrangement. In addition, this work has also introduced a novel approach based on hybrid Particle Swarm Optimization (PSO) and ACO (BWAS). The outcomes for the TSP arrangement show that the initial trail setup for the best particle can shorten the optimization’s accumulated process by a considerable amount. The mathematical test exhibition shows the proposed calculation’s viability over regular ACO- and PSO-ACO-based strategies. Jiang, C. [16] presents a hybrid algorithm called the ant colony-Parthenon genetic algorithm, provided by combining Parthenon genetic Algorithms (PGA) and the ant colony algorithm. The main idea of this work is to divide the variables into two parts. It exploits the PGA to comprehensively search for the best value of the first part variables and then utilizes the ACO to accurately determine the second part variables’ value. For comparative analysis, PGA, improved PGA (IPGA), Two-part Wolf Pack Search (TWPS), Artificial Bee Colony (ABC), and Invasive Weed Optimization (IWO) are adopted to solve MTSP and are validated with publicly available TSPLIB benchmarks. The comparative experiments show that the AC-PGA is sufficiently effective in solving large-scale MTSP and performs better than existing algorithms. All the above scholars have researched the combination of algorithms to improve the optimization ability. The results show that the method can improve the algorithm’s accuracy on a large scale. The effect of this technique combined with other algorithms is pronounced. The improved algorithm is generally better than the previous one but combining other algorithms will increase the algorithm’s complexity and reduce the convergence rate. In addition, we found that changing its pheromone updating mechanism can improve the algorithm’s performance. G. Liu [17] proposed a method to enhance pheromone update. The improved algorithm can update the pheromone dynamically and adaptively according to the density of the pheromone and the quality of the iterative optimal solution.

This paper optimizes the community distribution based on the above scholars’ research. UAVs are used to realize the “contactless distribution” of regional logistics. The logistics structure consists of a UAV, console, and customer. The distribution center arranges a drone to deliver daily life needs to customers with different epidemic conditions and needs in a particular area. The goal is to use drones to distribute supplies centrally. The console offers drones the shortest route through all customers considering the most critical time facto in unexpected situations. Each customer must be provided by drones once. This method can efficiently complete the logistics distribution to customers during the epidemic. Our previous research proposed a hybrid algorithm for solving the global shortest travel route problem by combining the ACA and the Physarum Polycephalum algorithm with Van der Waals forces. There are still areas for improvement in the solution accuracy of the path and the probability of reaching the optimal solution is low; however, adding Van der Waals forces has significantly improved the convergence speed by comparing the simulation results with the official optimal results. Therefore, this paper will improve the algorithm’s accuracy as the primary research goal. It is found that the ideas of crossover, variation, and conversion in the genetic algorithm are integrated into the optimal global solution of each iteration. The improvement can optimize the optimal global solution and improve the algorithm’s accuracy. This paper presents a hybrid algorithm based on the ant colony algorithm and Physarum Polycephalum algorithm. The core operator of the genetic algorithm is used to optimize the globally optimal path. The Van der Waals force is also added to disturb the pheromone updating process for UAV logistics distribution. This algorithm is superior in performance and optimization compared with other intelligent heuristic algorithms. An excellent result is presented in the problem of seeking the optimal route for UAV logistics distribution.

The structure of this chapter is as follows: In Section 2, the concepts and formulas of the NP-Hard problem, the Traveling Salesman Problem, the crossover of genetic algorithm, the mutation process idea, the basic concept, and the formula of Van der Waals forces, the ant colony algorithm, and the Physarum Polycephalum algorithm are given. Section 3 expounds on the basic framework of the hybrid algorithm combined with the genetic algorithm. In Section 4, simulation experiments are carried out on the official data set to compare with the mainstream algorithm and the efficiency of solving practical problems during the epidemic. Section 5 summarizes the full text.

## 2. Related Work

Section 2.1 describes the Traveling Salesman problem. The basic concept and formula of the related algorithms are shown in Section 2.2, Section 2.3, Section 2.4, Section 2.5.

### 2.1. Traveling Salesman Problem

The NP-HARD problems are problems where the solution to a problem can be proved or falsified in polynomial time. For example, any routing scheme can be calculated quickly in the backpacker problem. However, the optimal solution cannot be found in polynomial time. The Traveling Salesman Problem presented in this paper is an NP-hard problem.

The Traveling Salesman Problem is described as follows: A salesman wants to travel to several cities. He starts from one city and must pass through each city only once before returning to the starting city. The purpose is to find the shortest path, where “et. C” represents cities, and finally comes back after *n* cities; $C=\left\{c_{1},c_{2},c_{3}\ldots ,c_{n},c_{1}\right\}$. The objective function is:(1)$$L\left(C\right)=min\overset{n-1}{\underset{i=1}{\sum }}d\left(c_{i},c_{i+1}\right)+d\left(c_{n},c_{1}\right)$$
where *n* is the number of cities, $c_{i}$ is the number of cities, and $d\left(c_{i},c_{i+1}\right)$ represents the European distance from the city $c_{i}$ to $c_{i+1}$

### 2.2. Ant Colony Optimization

The ant colony algorithm is an efficient heuristic algorithm for solving discrete problems. The main idea is that ants leave a trail of volatile pheromones searching for a food source. After one round, subsequent ants tend to follow the path with higher pheromone levels. Over time, the pheromones of the other paths evaporate. Eventually, all the ants follow the path with the highest concentration of pheromones, resulting in the theoretically shortest path.

Each uses a roulette to select the probability of the next city, as shown in Formula (2):(2)$$P_{ij}^{k}\left(t\right)=\left\{\begin{array}{c} \frac{{\left[\tau_{ij}\left(t\right)\right]}^{\alpha }{\left[\eta_{ij}\left(t\right)\right]}^{\beta }}{\sum_{s\in S_{k}}{\left[\tau_{is}\left(t\right)\right]}^{\alpha }{\left[\tau_{is}\left(t\right)\right]}^{\beta }},\;\;\;\;j\in N_{k};\\ 0,\;\;\;\;\;\;\;\;\;\;\;\;\;\;\;\;otherwise; \end{array}\right.$$

Respectively, α and β represent the essential factors of pheromone concentration and the crucial factors of heuristic information; $\tau_{ij}^{k}$ represents the pheromone concentration from city *i* to city *j*; and $\eta_{ij}$ represents the important heuristic factors from city *i* to city *j.*

When all ants go through one round, the ant colony system updates the pheromone updating strategy formula as shown in (3):(3)$$\tau_{ij}^{k}=\left(1-\rho \right)\tau_{ij}^{k}+\sum_{k=1}^{m}{\left(L_{k}\right)}^{-1}$$
where ρ is the pheromone volatilization coefficient.

### 2.3. Physarum Polycephalum Algorithm

The principle of the Physarum Polycephalum algorithm (PPA) is the process of foraging by the Physarum Polycephalum. The food source is regarded as the endpoint, and the flow and conductivity of the pipe will present a positive feedback mechanism. The flow of the pipe will increase with the thickening of the pipe. The corresponding increase in flow rate will also affect the radius of the pipeline. A thicker tube will appear and then increase the flow by adding more nutrients to this tube when the Physarum Polycephalum is looking for a food source according to the positive feedback mechanism. Finally, a theoretical shortest pipeline is generated.

The flow formula of the Physarum Polycephalum algorithm is shown in (4)
(4)$$Q_{ij}=\frac{D_{ij}}{d_{ij}}\left(p_{i}-p_{j}\right)$$

$D_{ij}$ represents the conductivity between pipes, $d_{ij}$ is the distance between pipe *i* and *j*, and ($p_{i}-p_{j}$) represents the pressure difference between points *i* and *j*.

According to Kirchhoff’s theorem, pheromone flow is equal from the inlet and outlet, which can be expressed by Formula (5):(5)$$\overset{n}{\underset{j=1,j\neq i}{\sum }}q_{ij}=\overset{n}{\underset{j=1,j\neq i}{\sum }}q_{ij}\frac{d_{ij}\left(t\right)}{l_{ij}}\left(p_{i}\left(t\right)-p_{j}\left(t\right)\right)=\left\{\begin{array}{l} flow,\;\;i=\sigma \\ -flow,\;\;i=\tau \\ 0,\;\;\;\;\;\;\;\;else \end{array}\right.$$

The formula of conductivity $D_{ij}$ changing with time is as follows:(6)$$\frac{dD_{ij}}{d_{t}}=f\left(\left|Q_{ij}\right|\right)-rD_{ij}$$
where *r* is the decline rate of the pipeline and $f\left(\left|Q_{ij}\right|\right)$ is a monotone increasing function. The path planning problem can be expressed in formula (7) as:(7)$$f\left(Q\right)=\left|Q_{ij}\right|∕\left(1+\left|Q_{ij}\right|\right)$$

*R* = 1 is selected as the pipeline decline rate, and Formula (8) can be simplified as:(8)$$\frac{D_{ij}^{n+1}-D_{ij}^{n}}{\delta t}=\frac{\left|Q_{ij}\right|}{1+\left|Q_{ij}\right|}-D_{ij}^{n+1}$$

According to the Hagen Poiseuille equation, the flow formula of the pipeline between two nodes is as follows:(9)$$\begin{array}{c} d_{ij}=\frac{\pi r_{ij}^{4}}{8\zeta l_{ij}} \end{array}$$

In Formula (9), $r_{ij}$ is the pipe’s radius, $l_{ij}$ is the pipe’s length, and ζ is the fluid’s viscosity.

### 2.4. The Van der Waals Force

The Van der Waals force is a molecular force. The size of the Van der Waals force is proportional to the size of the molecule. The primary sources are divided into the following three mechanisms:

**Orientation force** The interaction between polar molecules;

**Inducing force** Polar molecules polarize nonpolar molecules, producing induced dipole moments, which interact with each other;

**Dispersion force** The probabilistic motion of a pair of nonpolar molecules over their electrons can work together to produce a pair of opposing instantaneous dipole moments that interact.

The calculation formula is as follows for the Van der Waals force between two molecules:(10)$$F-\frac{A}{d^{a}}+\frac{B}{d^{b}}$$

The values of *A* and *B* come from the molecules themselves; *d* is the distance between the molecules; *a* is usually 12 and *b* is 6; and the front part of this formula is the repulsion between the molecules. The back part of this formula is the attraction between the molecules.

### 2.5. Genetic Algorithm

The genetic algorithm’s principle is to encode chromosomes and then cross and mutate the offspring. Finally, the new offspring will be compared with the original offspring. Suppose it is better than the original offspring. In that case, the original offspring will be replaced, and the process will be repeated until the most theoretical path is found.

Crossover refers to swapping chromosome segments to produce two new offspring, then counting the length of the new offspring and replacing them if they are better than the original. In a typical single-point crossover, two individuals are randomly selected to cross and produce new offspring in the following way:

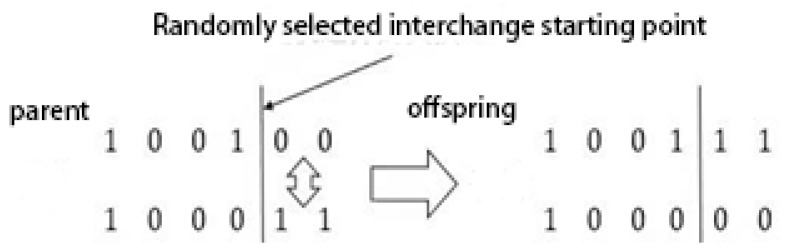


A mutation is a mutation in a gene segment or a gene point on a chromosome. One chromosome is randomly replaced with other values and then compared to the original, eliminating the poor one. The single-point mutation is shown as follows:

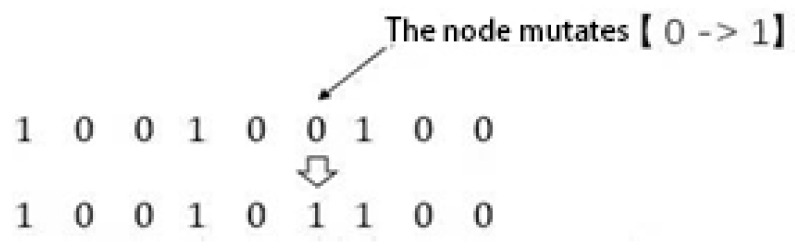


## 3. Genetic Algorithm Improved the Hybrid Algorithm with the Van der Waals Force

A hybrid algorithm based on the Physarum Polycephalum and ant colony algorithms mixed with the Van der Waals force is present in our previous studies. The algorithm significantly optimized the algorithm convergence speed and the optimal solution to a certain extent but the optimization effect is not high. This article mainly focuses on the optimization results in which the crossover and mutation operations of genetic algorithms are introduced to the process of optimal path optimization.

### The Van der Waals Force on Physarum Polycephalum Algorithm Based on the Genetic Algorithm Mix Ant Colony Optimization

Our improved method’s effect is adjusting the pheromone updating mechanism by adding factors such as flow rate and conductivity and further optimizing pheromone updating by using a positive feedback mechanism. The pheromone updating formula of the hybrid algorithm is shown in (11):(11)$$\tau_{ij}^{k}=(1-p)\tau_{ij}^{k}+\overset{m}{\underset{k=1}{\sum }}{\left(L_{k}\right)}^{-1}+\lambda e^{-{\frac{1}{Tmax}}^{2}\frac{Q_{ij}M}{I_{0}}}$$
where λ∈0−1; t is the current iteration number; $T_{max}$ is the maximum iteration number; *M* = $n^{2}$; and *n* is the number of cities. After the Van der Waals force is added, the pheromone updated formula is shown in (12):(12)$$\tau_{\mathrm{ij}}^{k}=(1-p)\tau_{\mathrm{ij}}^{k}+\sum_{k=1}^{m}{\left(L_{k}\right)}^{-1}+{\lambda e}^{-{\frac{1}{\mathrm{Tmax}}}^{2}\frac{Q_{\mathrm{ij}}M}{I_{0}}}+F=\;(1-p)\tau_{\mathrm{ij}}^{k}+\overset{m}{\underset{k=1}{\sum }}{\left(L_{k}\right)}^{-1}+{\lambda e}^{-{\frac{1}{\mathrm{Tmax}}}^{2}\frac{Q_{\mathrm{ij}}M}{I_{0}}}-\frac{A}{d^{a}}+\frac{B}{d^{b}}$$
where *a* > *b* and the values of *A*, *B*, *a*, *b* should be changed with the optimal distance in each pheromone iteration. The corresponding values of *A* and *B* should also be smaller when the distance is too small. Otherwise, the pheromone value of each time will be too high and directly fall into the optimal local solution of the program. It is the same if the distance is too large. Therefore, *A, B*, *a*, *b* should be adjusted for different data.

On this basis, this paper introduces the path-searching process to optimize the global optimal. The crossover rate and mutation rate are set under the condition of meeting the crossover rate to make the globally optimal path cross-processing. Then, the Euclidean distance is calculated using the original path with contrast. One point in the path is mutated when the mutation rate is satisfied. The original node of the mutation is exchanged with the node of the mutation point. After the new path is obtained, it is compared with the original path again. Finally, the maximum number of iterations is reached to end the algorithm. The optimal global solution is obtained.

The pseudocode of the improved Algorithm 1 named VP-GA-ACO (The Van der Waals force on Physarum Polycephalum algorithm based on the genetic Algorithm mix ant colony optimization) is as follows:


**Algorithm 1: VP-GA-ACO**
   Input: Multiobjective traveling salesman problem with two weights on each edge   Output: Pareto front   (**i**) **Initialization process**   Step 1 Initialize parameter and variable values used in the algorithm;   Step 2 A model was established according to the positive feedback mechanism of Physarum Polycephalum. The model was used to roughly solve the target problem of the traveling agent problem and obtain the pheromone matrix. Then obtain the initialization matrix of the mixed algorithm by using initializing the pheromone matrix of the ant colony algorithm;   Step 3 Set the iteration counter to *n* = 0.   (**ii**) **Iterative process**   Step 4 While (*n* < Tsteps), Do//Tsteps indicates the total number of iterations;   Step 5 The ant colony selects paths according to the pheromone matrix and heuristic function;   Step 6 Record the cities that each ant passes through in turn in the taboos table and find the Pareto front of this iteration by comparison;   Step 7 Update the pheromone matrix according to the pheromone updating rules of the ant colony algorithm optimized by the Van der Waals force and Physarum Polycephalum algorithm;   Step 8 Set the crossover probability and mutation probability based on the genetic algorithm. If the random number is between the crossover probability and mutation probability, the local path will be crossed or mutated to obtain a better path and replace the current longer path;   Step 9 Iterate the counter *n* = *n* + 1;   End.   (**iii**) **Results obtained**   Step 10 output the Pareto front.   The end.

## 4. Experimental Research and Analysis

In this paper, using the Python platform to perform the desires of a Pareto frontier graph simulation experiment, through the experiments on the prototype validating the effectiveness of the algorithm itself, and then through the study of the logistics distribution of Changchun Pearl Village work, we received results that verify the algorithm has more excellent robustness.

Python 3.10 was used to conduct simulation experiments on five official data sets. The running results of the VP-GA-ACO algorithm are shown as follows (Figure 1, Figure 2, Figure 3, Figure 4 and Figure 5).

The above figures show the operating results of five data sets using the VP-GA-ACO algorithm. The data set calculated by the algorithm is shown in Table 1. In Table 1, VPACO stands for the Van der Waals force on the Physarum Polycephalum algorithm based on ant colony optimization and MMAS stands for the maximum and minimum ant colony algorithm.

The *Opt* value in Table 1 is obtained from TSPLIB; *Alg*. represents the algorithm; $S_{min}$ represents the optimal solution of the algorithm operation; $S_{\mathrm{average}}$ represents the average of the results of 500 iterations of the algorithm; *Gap* = $\frac{S_{\min}-S_{\mathrm{Opt}}}{S_{\min}}\times 100$, represents the deviation rate between the minimum value and the optimal solution of the algorithm simulation; the smaller the *Gap* value, the better the running result. *AVR* = $\left|\frac{S_{\mathrm{average}}-S_{\mathrm{Opt}}}{S_{\mathrm{Op}}}\right|\times 100$, *AVR* represents the deviation rate between the mean of the algorithm simulation data and the optimal solution. The smaller the value of *AVR*, the faster the algorithm converges.

Below are the iterated graphs of the five data algorithms, as shown in Figure 6, Figure 7, Figure 8, Figure 9, Figure 10.

It can be concluded from the results of curve iteration that the VP-GA-ACO algorithm is better than other algorithms and it has a better ability to find the optimal solution and the fastest speed to force the optimal solution. It is proved that the algorithm has strong stability, high solving speed, and strong robustness. The VP-GA-ACO algorithm is compared with other mainstream algorithms. The comparison results are shown in Table 2:

The comparison results show that the VP-GA-ACO algorithm is superior to current mainstream algorithms for data sets with 50–150 nodes. Its accuracy is also higher than other mainstream algorithms. We applied this algorithm to the logistics distribution route planning in the Pearl Residential Area of Changchun during the COVID-19 epidemic. The UAV control center inputs user location information and used the VP-GA-ACO algorithm in the control center to complete the route selection. The UAV distribution process is shown in Figure 11.

The UAV control center input user coordinates and used the VP-GA-ACO algorithm to generate the optimal distribution scheme, which was provided to the UAV platform. The UAV platform instructed the UAV to implement “contactless distribution” for users in the community. Finally, the UAV returned to the UAV platform. In this process, there is no human contact, avoiding the spread of the virus caused by direct or indirect contact between people and completing the “last mile” distribution of “no contact.”

In order to prove the practicability of the algorithm, this paper will use the coordinate position of the user’s building in the Pearl District of Changchun city to simulate the UAV logistics distribution experiment. For further investigation, the Mingzhu community in Changchun city includes 43 buildings in Area B, 66 in Area C, and 38 buildings in Area D, totaling 147 buildings. Among them is the rookie station as a UAV dispatch center. The building distribution of Changchun Mingzhu Residential District is shown in Figure 12.

The simulation experiment was divided into three, and the data were randomly taken out. The number of floors to be distributed to was defined as 40, 60, and 80. The floors to be distributed in the three tests are shown in Table 3, excluding the coordinates of the dispatching center.

The total flight distance of the UAV determined by the first test result diagram is 4.2558 KM, as shown in Figure 13. The flight plan is: 1-41-27-31-28-33-35-32-34 and 37-39-40-38-36-26-25 to 30-29-15-16-14-2-3-5-4-6-5-4-6-14-2-3-19-11-13-19-11-13-24-21-20-1; where 1 is the dispatching center; 2–41, respectively, represent the building numbers corresponding to the first test.

The total flight distance of the UAV determined by the second test result diagram is 4.9410 km, as shown in Figure 14. The flight plan is: 1-41-35 to 37-40-39-38-36-27-28-29-11-13-14 to 15-17-29-11-13-6-20-18-4-5-2-19-4-5-2-6-20-18-22-46-47-48-24-26-25-58-59-61-60 49-51-50-57-56-44-55 to 45-54-53-52-43-42-33-32-31-30-34-1; where 1 is the dispatching center; and 2-61, respectively, represent the building numbers corresponding to the second test.

The total flight distance of the UAV determined by the third test result diagram is 5.4081 km. The flight plan is: 1-48-50-54-49-60-61-55-51-62-63-64-75-76-65-77-79-78-58-57-74-59-73-53-52-80-56-81-72-71-35 to 37-38-36-34-70-69-68-67-33-6 6-32-29-31-30-15-16-17-14-13-12-11-10-9-28-10-9-28-8-3-4-13-12-11-39-40 and 42-18-26-20-22-19-21-23-24-25-45-44-46-43-47-1; No. 1 is the dispatching center, and No. 2–81, respectively, represent the building numbers corresponding to the third test. The experiment proves that the algorithm has a good effect on contactless distribution in the community during the epidemic.

## 5. Conclusions

This paper presents an implementation with “no contact distribution” to provide corresponding research during the critical situation. The UAV is used as the main realization tool in the scheme. The crossover and mutation in the genetic algorithm were applied to the globally optimal path with Van der Waals force optimization based on the original results. Finally, simulation experiments confirmed that the algorithm’s accuracy improved compared with the previous VPACO algorithm in solving the optimal solution. We applied the algorithm to the contactless logistics distribution direction of the Changchun Mingzhu Residential area during the epidemic. We verified that the algorithm was effective and efficient. We will research the supply distribution problem in emergency situations, and our research will focus on designing a rapid and integrated mode of transportation to solve complex transportation problems effectively.

## Figures and Tables

**Figure 1 biomimetics-08-00096-f001:**
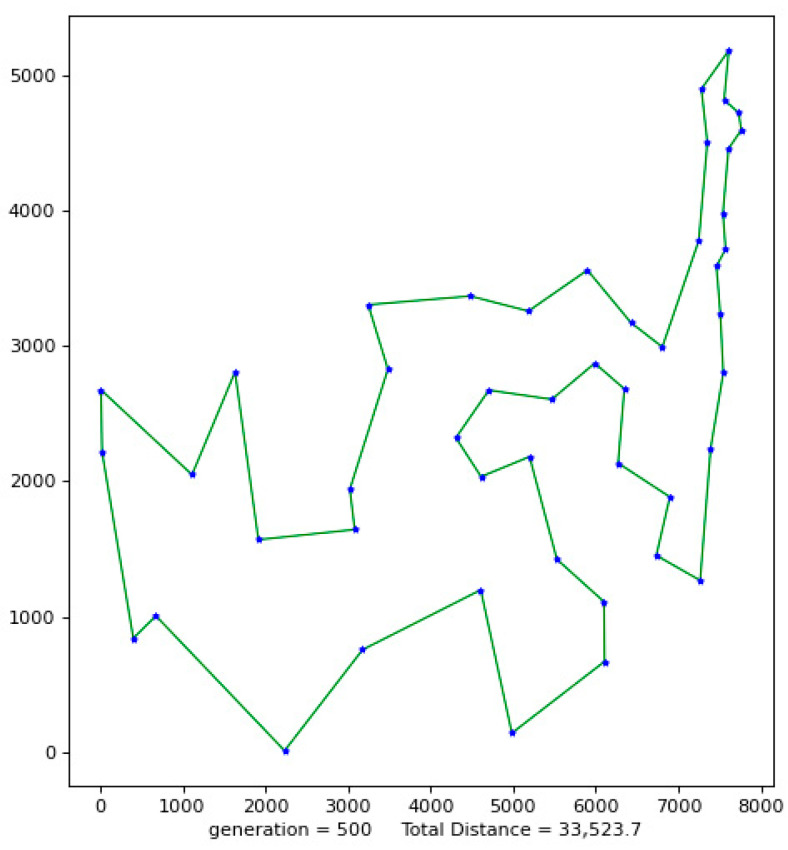
The best result of Att48.

**Figure 2 biomimetics-08-00096-f002:**
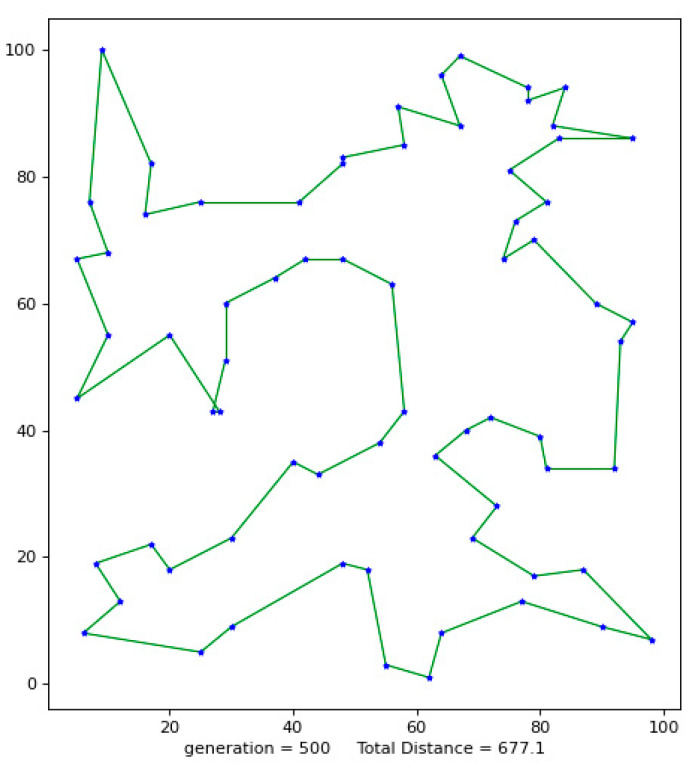
The best result of St70.

**Figure 3 biomimetics-08-00096-f003:**
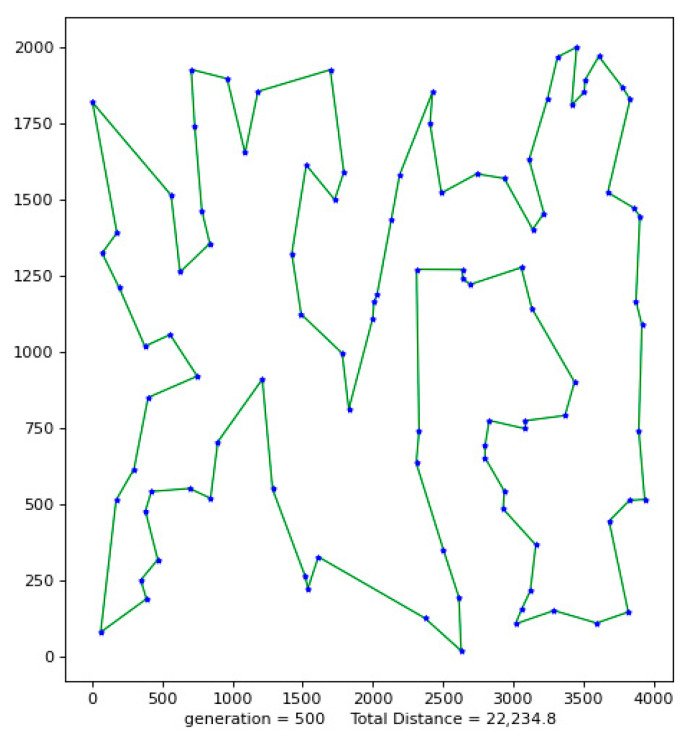
The best result of KroB100.

**Figure 4 biomimetics-08-00096-f004:**
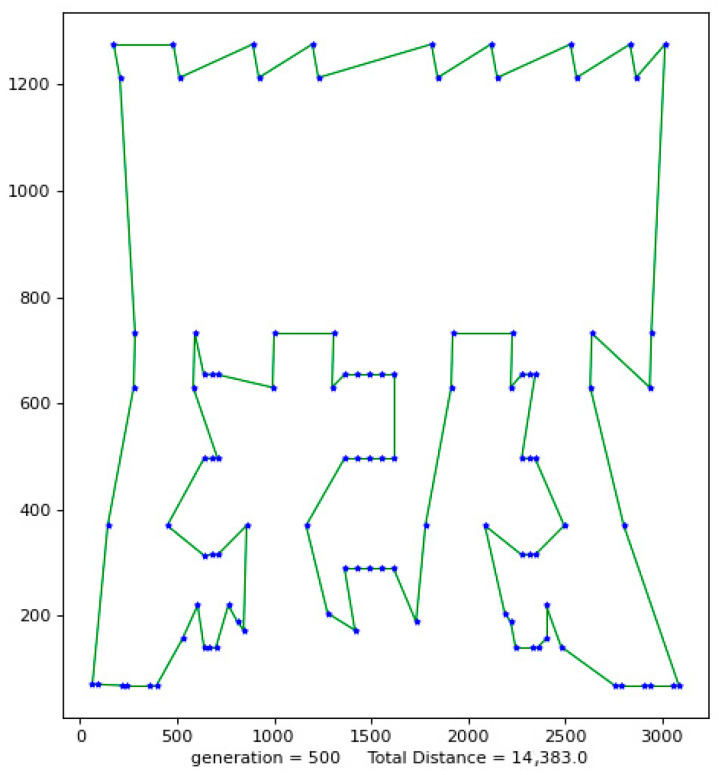
The best result of Lin105.

**Figure 5 biomimetics-08-00096-f005:**
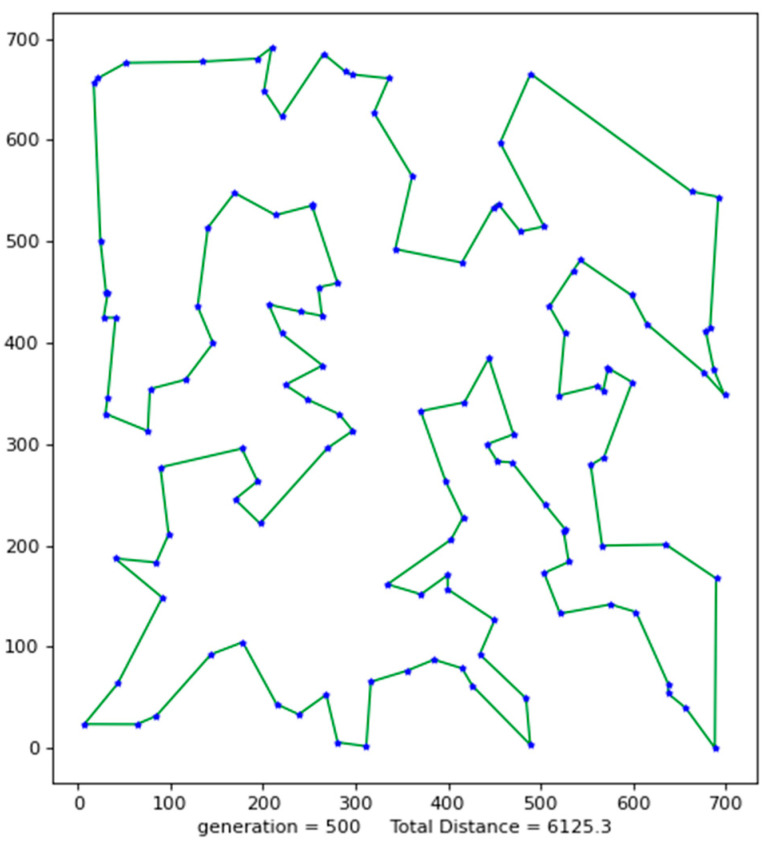
The best result of Ch130.

**Figure 6 biomimetics-08-00096-f006:**
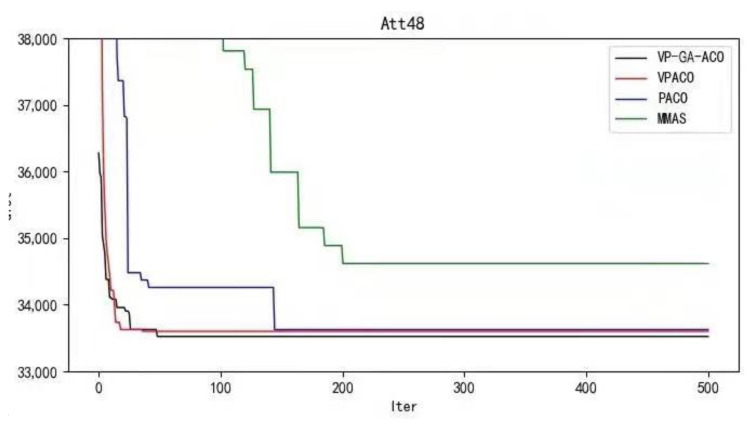
Iteration curves of ATT48.

**Figure 7 biomimetics-08-00096-f007:**
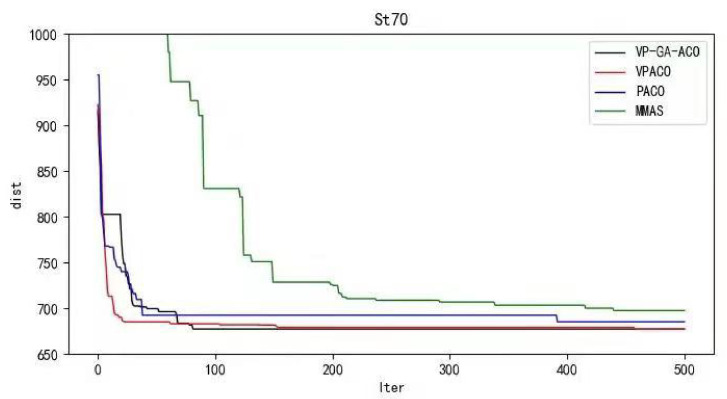
Iteration curves of ST70.

**Figure 8 biomimetics-08-00096-f008:**
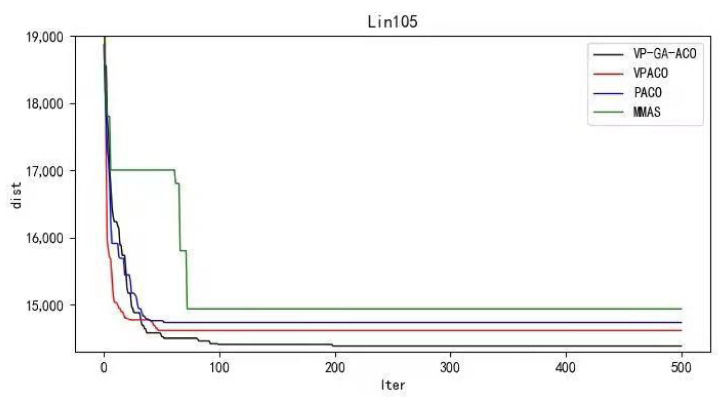
Iteration curves of Lin105.

**Figure 9 biomimetics-08-00096-f009:**
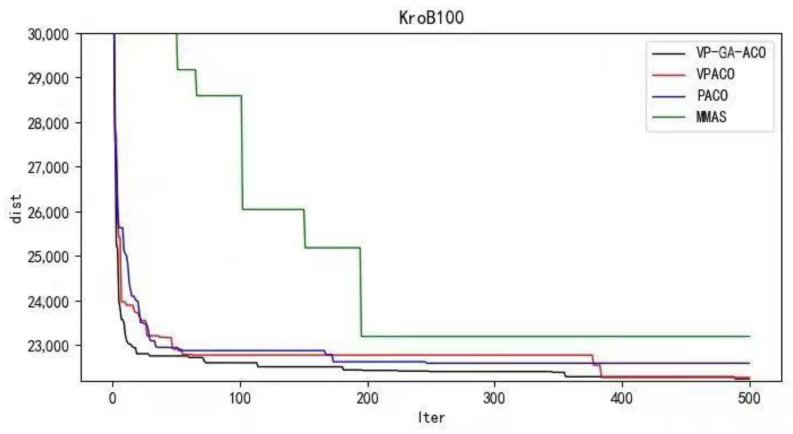
Iteration curves of KroB100.

**Figure 10 biomimetics-08-00096-f010:**
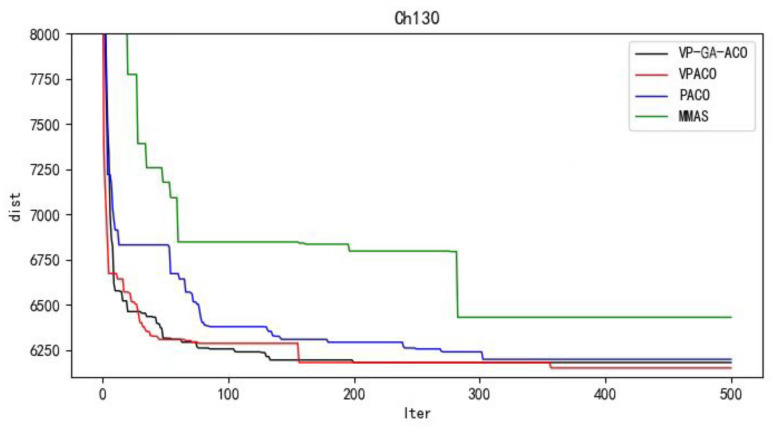
Iterative curves of Ch130.

**Figure 11 biomimetics-08-00096-f011:**
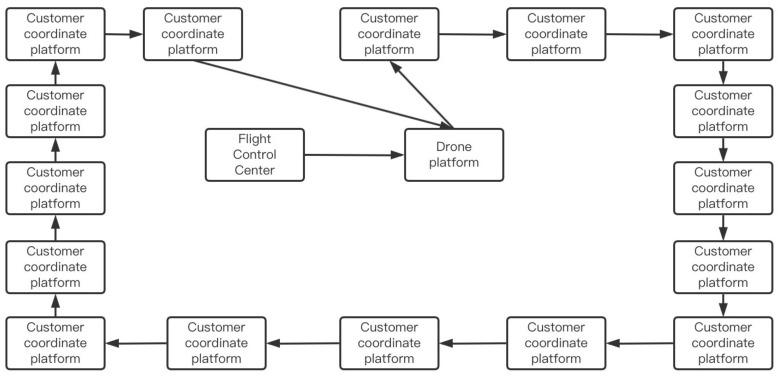
UAV logistics distribution process demonstration diagram.

**Figure 12 biomimetics-08-00096-f012:**
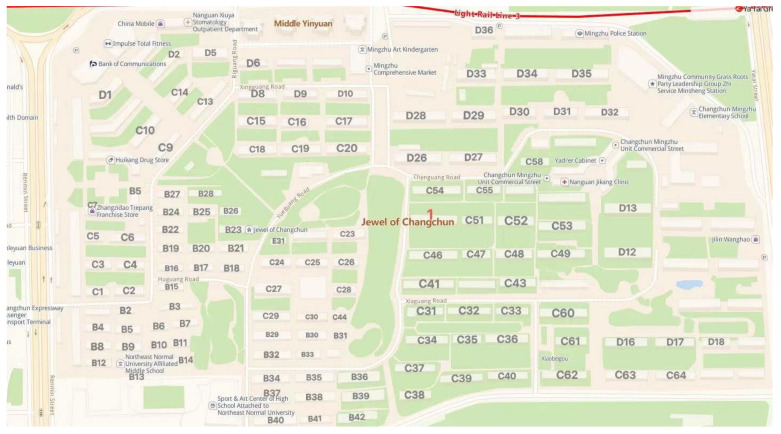
Building distribution of Mingzhu District in Changchun.

**Figure 13 biomimetics-08-00096-f013:**
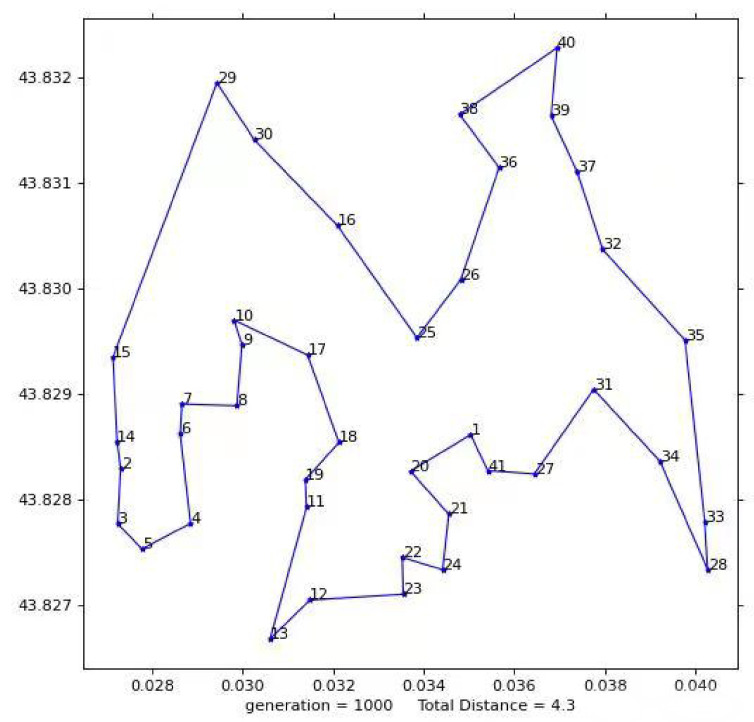
First test results.

**Figure 14 biomimetics-08-00096-f014:**
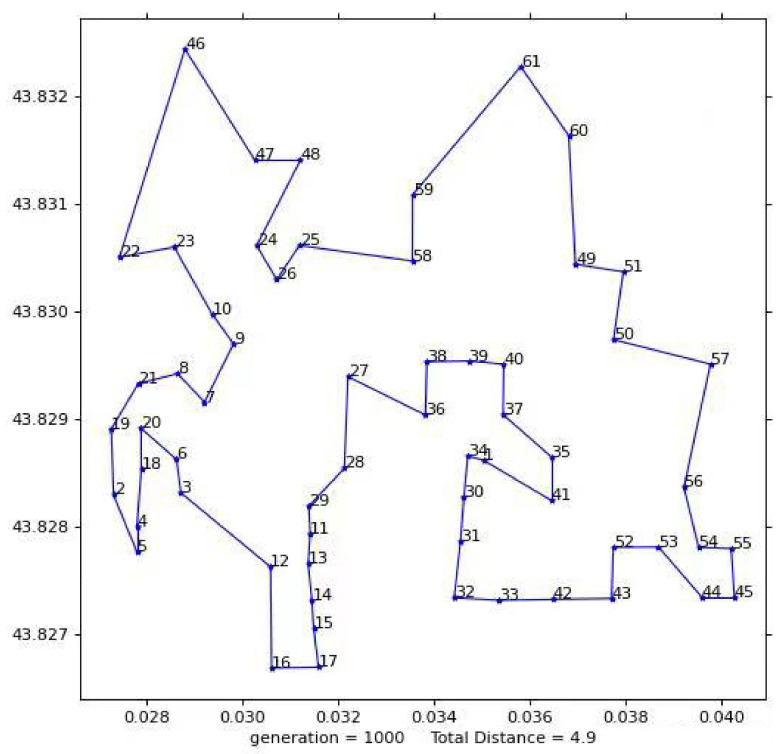
The second test results.

**Table 1 biomimetics-08-00096-t001:** Improved algorithm and existing algorithm data test results.

Instance	Opt	Alg.	$S_{\min}$	$S_{\mathrm{average}}$	Gap	AVR
Att48	33,523	VP-GA-ACO VPACO PACO MMAS	33,523 33,600 33,625 34,355	33,572 33,676 34,070 34,446	0 0.20 0.27 2.5	0.12 0.43 1.62 2.73
St70	675	VP-GA-ACO VPACO PACO MMAS	677 677 685 697	683 680 702 723	0.29 0.29 1.48 3.26	1.19 0.74 4.00 7.11
KroB100	22,141	VP-GA-ACO VPACO PACO MMAS	22,244 22,277 22,597 23,198	22,356 22,412 24,233 25,124	0.47 0.61 2.06 4.77	0.97 1.22 9.44 13.47
Lin105	14,383	VP-GA-ACO VPACO PACO MMAS	14,383 14,615 14,733 14,938	14,413 14,738 15,116 15,347	0 1.61 2.43 3.86	0.21 2.47 5.10 6.70
Ch130	6110	VP-GA-ACO VPACO PACO MMAS	6180 6150 6198 6430	6307 6210 6395 6870	1.15 0.65 1.44 5.24	3.22 1.64 4.66 12.44

**Table 2 biomimetics-08-00096-t002:** The VP-GA-ACO algorithm compared with other mainstream algorithms.

Test	Opt	VP-GA-ACO	GMHSA [18]	IST [19]	*CSO* [20]	*IFA* [21]
Eil51	429	429	-	429	447	429
St70	675	677	-	677	694	683
KroA100	21,282	21,298	34376	21,285	-	-
Lin105	14,383	14,383	-	14,406	-	-
Ch130	6110	6180	6671	-	-	-

**Table 3 biomimetics-08-00096-t003:** Test UAV logistics and distribution building.

First Test Building (40)	B01 B08 B11 B13 B15 B16 B18 B23 B26 B30 B38 B40 C01 C05 C20 C22 C28 C30 C31 C33 C35 C37 C38 C39 C50 C55 C60 C66 D05 D08 D12 D15 D19 D20 D25 D30 D32 D33 D35 D38
Second Test Building (60)	B01 B03 B05 B09 B15 B20 B22 B26 B28 B30 B32 B33 B35 B38 B40 B41 C02 C03 C04 C06 C08 C09 C18 C19 C21 C23 C28 C30 C32 C35 C39 C40 C42 C45 C46 C48C50 C51 C52 C60 C62 C63 C65 C66 D03 D08 D09 D11 D13 D15 D16 D17 D18 D19 D20 D25 D26 D28 D35 D37
Third Test Building (80)	B04 B05 B06 B08 B10 B11 B13 B15 B16 B19 B20 B21 B23 B24 B25 B28 B30 B34 B35 B37 B38 B40 B41 B42 B43 C03 C04 C08 C09 C10 C11 C13 C15 C16 C18 C19 C21 C22 C28 C29 C30 C35 C37 C38 C39 C42 C43 C45 C48 C49 C50 C51 C52 C53C55 C56 C57 C59 C60 C61 C62 C63 C64 C66 D01 D03 D04 D05 D08 D09 D10 D12 D15 D17 D18 D19 D22 D25 D26 D28

## Data Availability

The data that support the findings of this study are available on request from the corresponding author, [whdzy2000@vip.sina.com], upon reasonable request.

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
