# Peer review of "Application of Hybrid Swarming Algorithm on a UAV Regional Logistics Distribution"

_biomimetics, 2023, doi:10.3390/biomimetics8010096_

Round 1
Reviewer 1 Report
-
The abstract is too short.
-
The introduction does not provide sufficient background: the authors do not provide references to other (non-meta-heuristic) approaches to solving the TSP; the authors represent existing ACO-based hybrid methods using only a few articles (references 8-12, lines 42-68); the description of the TSP for UAV routes planning is too short (lines 74-95).
-
Since the authors use four algorithms in one algorithm, two of which (GA and ACO) are complex in themselves, the description of the “Algorithm: VP-EO-ACO” should be made much more detailed. For example, Step 2 and 8.
-
The authors write: “set the running times to no more than 10 times for each simulation data and select the best one of the 10 times as the result“ (lines 259-260). I think that 10 runs may not be enough to draw conclusions. In addition, please add the average value and standard deviation.
-
What are the meanings of VPACO, PACA, MMAS in Table 4.1?
-
It is recommended to show in the results how each addition influenced separately. For example, compare VP-EO-ACO,
-
Please translate the Figure 4.11 into English.
-
Figure 4.15 is omitted.
Author Response
Re: Manuscript ID: biomimetics-2182064 and Title: Application a hybrid swarming algorithm on UAV regional logistics distribution
Thank you for the reviewers’ comments concerning our manuscript entitled “Application of a hybrid swarming algorithm on UAV regional logistics distribution” (biomimetics-2182064). Those comments are valuable and very helpful. We have read through the comments carefully and have made corrections. We uploaded the file of the revised manuscript based on the instructions provided in your letter. Revisions in the text are shown using the blue highlight for addition. The responses to the reviewer's comments presented following.
As a master’s student, this is my first time submitting a paper to biomimetics. I am very grateful to thank you for allowing us to resubmit a revised copy of the manuscript and we highly appreciate your time and consideration.
Sincerely.
Hongda Yu Yi Zhang.
Q1. The abstract is too short.
Response: The optimization mechanism is added in the abstract, and the work contribution is increased. (lines 8,12-15)
Q2. The introduction does not provide sufficient background: the authors do not provide references to other (non-meta-heuristic) approaches to solving the TSP; the authors represent existing ACO-based hybrid methods using only a few articles (references 8-12, lines 42-68); the description of the TSP for UAV routes planning is too short (lines 74-95).
Response: We added background (lines 20-31), essential method(Lines 47-56), comparison with other heuristics(lines 65-74), add ACO-based hybrid methods articles(lines 91-114) ,add the description improve method for UAV routes planning(lines 125-148)
Q3. Since the authors use four algorithms in one algorithm, two of which (GA and ACO) are complex in themselves, the description of the “Algorithm: VP-EO-ACO” should be made much more detailed. For example, Step 2 and 8.
Response: We add the detailed description of Step 2 and 8(lines 285)
Q4. The authors write: “set the running times to no more than 10 times for each simulation data and select the best one of the 10 times as the result“ (lines 259-260). I think that 10 runs may not be enough to draw conclusions. In addition, please add the average value and standard deviation..
Response:We improved the experiment and added meaningful results according to the reviewer's suggestions(lines 308-318)
Q5. What are the meanings of VPACO, PACA, MMAS in Table 4.1?
Response :This has been clarified in the revised version of the manuscript(lines 304-306)
Q6. It is recommended to show in the results how each addition influenced separately. For example, compare VP-EO-ACO,
Response :This has been clarified in the revised version of the manuscript (lines 310-317)
Q7. Please translate the Figure 4.11 into English.
Response :This has been clarified in the revised version of the manuscript, and Figure 4.11 has been translate .
Q8. Figure 4.15 is omitted.
Response :omitted.

Author Response
Re: Manuscript ID: biomimetics-2182064 and Title: Application a hybrid swarming algorithm on UAV regional logistics distribution
Thank you for the reviewers’ comments concerning our manuscript entitled “Application of a hybrid swarming algorithm on UAV regional logistics distribution” (biomimetics-2182064). Those comments are valuable and very helpful. We have read through the comments carefully and have made corrections. We uploaded the file of the revised manuscript based on the instructions provided in your letter. Revisions in the text are shown using the blue highlight for addition. The responses to the reviewer's comments presented following.
As a master’s student, this is my first time submitting a paper to biomimetics. I am very grateful to thank you for allowing us to resubmit a revised copy of the manuscript and we highly appreciate your time and consideration.
Sincerely.
Hongda Yu Yi Zhang.
Q1.The abstract is not compact. The rationale of this study and the contribution of this paper are not detailed described.
Response:The optimization mechanism is added in the abstract, and the work contribution is increased. (lines 8,12-15)
Q2.The literature survey is not exhaustive. Many recent works are not accommodated in the survey.
Response:We add ACO-based hybrid methods articles(lines 91-114)
Q3. The motivation of this research is not well established. The rationale of using the proposed approach is not clear.
Response: We add the description improve method for UAV routes planning(lines 125-148)
Q4. The contributions of the paper are not properly advocated in Introduction section.
Response:We added background (lines 20-31), essential method(Lines 47-56), comparison with other heuristics(lines 65-74),
Q5. In numerical analysis a comparative study is needed in which showing the proposed methods and the traditional Traveling Salesman 33 Problem (TSP).
Response:We improved the experiment and added meaningful results according to the reviewer's suggestions(lines 308-318)

Round 2
Reviewer 1 Report
The manuscript has been significantly improved. It can be accepted
Reviewer 2 Report
The authors have revised the paper with respect to the comments of the reviewers. I suggest that the paper be accepted at the present from.